# Sociodemographic and geographic determinants of childhood immunization coverage and equity in Ghana: Analysis of the 2022 demographic and health survey

Ahlam Tunteeya Saani[1]*, Goldfield Edem Azumah[2]

1 University of Ghana Dental School, College of Health Sciences, University of Ghana, Korle Bu, Accra, Ghana, 2 Korle Bu Teaching Hospital, Accra, Ghana

* ahlamsaani@icloud.com

## Abstract

### Background

Childhood immunization remains a cornerstone of public health in sub-Saharan Africa, yet substantial gaps in coverage, equity, and schedule adherence persist. This study examines immunization coverage, sociodemographic disparities, predictors of completion, and policy implementation effectiveness in Ghana using nationally representative data.

### Methods

Data from 3,788 children aged 12–35 months in the 2022 Ghana Demographic and Health Survey were analysed using stratified two-stage cluster sampling across 614 enumeration areas. Weighted coverage estimates were calculated for individual vaccines and composite indicators including full immunization, dropout rates, and timeliness. Design-adjusted chi-square tests identified bivariate associations between sociodemographic factors and immunization outcomes. Multivariate logistic regression models incorporated sociodemographic characteristics, healthcare utilization, and information access to identify independent predictors. Equity analysis employed concentration indices, rate ratios, and intersectional stratification.

### Findings

Full immunization coverage reached 69.5% (95% CI: 67.3 to 71.7%), with 25.7% partial immunization and 4.8% zero-dose children. Substantial inequalities emerged across wealth (17.4 percentage point gap), maternal education (18.3 percentage point gap), and region (35.7 percentage point gap). The concentration index of 0.045 indicated modest pro-rich inequality. Dropout rates were concerning, particularly 28.9% attrition between measles-rubella first and second doses among age-eligible

**Data availability statement:** The data underlying this study are publicly available from the Demographic and Health Surveys (DHS) Program. Researchers can access the Ghana 2022 DHS dataset by registering at https://dhsprogram.com and submitting a data access request. Contact: The DHS Program, ICF International, 530 Gaither Road, Suite 500, Rockville, MD 20850, USA (info@dhsprogram. com).

**Funding:** The author(s) received no specific funding for this work.

**Competing interests:** The authors have declared that no competing interests exist.

children. In adjusted models, facility delivery (aOR=1.43, 95% CI: 1.08 to 1.90), four or more antenatal care visits (aOR=1.55, 95% CI: 1.19 to 2.01), higher wealth (aOR=1.39, 95% CI: 1.04 to 1.87), and maternal secondary or higher education (aOR=1.34, 95% CI: 1.04 to 1.72) independently predicted full immunization.

## Interpretation

Ghana's immunization program achieves broad initial access but faces critical challenges in retention, timeliness, and equity. Policy priorities should emphasize completion-focused interventions including tracking systems, reminder mechanisms, and targeted outreach to disadvantaged populations.

## Introduction

Vaccination stands as one of public health's most effective interventions, having prevented millions of deaths and nearly eradicated diseases. [1] However, substantial gaps persist in achieving universal coverage, ensuring schedule completion, reaching disadvantaged populations equitably, and administering vaccines at age-appropriate intervals that optimize protection during vulnerable periods. [2,3]

Sub-Saharan Africa faces challenges in immunization delivery despite sustained investments and policy commitments. [4] The region accounts for a disproportionate share of unimmunized and under-immunized children globally, with coverage plateaus, periods where immunization rates stagnate despite continued programmatic investment, and recent reversals threatening progress toward Sustainable Development Goal targets and the Immunization Agenda 2030 objectives. [4] Persistent inequalities in coverage across socioeconomic, geographic, and demographic dimensions indicate that universal policy frameworks have not translated into equitable service delivery. [5] Dropout rates between vaccine doses remain elevated, timeliness of administration frequently deviates from recommended schedules, and health system weaknesses including stock-outs, inadequate cold chain capacity, and insufficient healthcare worker training compromise programme effectiveness. [6]

Ghana has made substantial progress in childhood immunization over the past two decades, with coverage for traditional vaccines increasing from below 60% in the early 2000s to above 80% for most antigens by the 2020s. [7] The country has successfully introduced multiple newer vaccines including pneumococcal conjugate vaccine in 2012, rotavirus vaccine in 2012, and measles-rubella second dose in 2018, demonstrating programmatic capacity for innovation. [7] Ghana's Expanded Programme on Immunization operates through an integrated health service delivery platform combining facility-based services with outreach activities, supported by the Ghana Health Service district health management structures. [7]

Despite Ghana's progress, substantial knowledge gaps remain regarding the patterns, determinants, and equity dimensions of immunization coverage using population-based data that can validate administrative statistics. Previous analyses have often focused on single dimensions such as urban-rural disparities or

wealth-related inequalities in isolation, without examining how multiple disadvantages intersect to create particularly vulnerable populations. [7,8–11] Limited attention has been given to retention through multi-dose series and across the complete schedule, with most analyses emphasizing initiation rather than completion. Timeliness of vaccination has received inadequate attention in Ghana and broader sub-Saharan African contexts due to data quality challenges with vaccination dates.

This study addresses these knowledge gaps through comprehensive analysis of immunization coverage, disparities, determinants, and policy implementation using data from the 2022 Ghana Demographic and Health Survey. The research objectives encompass documenting coverage patterns for individual vaccines and composite indicators including full immunization, partial immunization, and zero-dose prevalence, while examining timeliness of vaccine administration relative to recommended schedules. The analysis identifies disparities across sociodemographic characteristics including residence, wealth, maternal education, region, and intersectional categories, and determines independent predictors of immunization completion through multivariate regression. The study assesses equity using multiple complementary inequality measures, analyzes dropout patterns identifying critical retention challenges, and evaluates policy implementation effectiveness against national targets.

## Materials and methods

### Study design and data source

This study analysed data from the 2022 Ghana Demographic and Health Survey, a nationally representative cross-sectional household survey conducted between October 17, 2022, and January 14, 2023, by the Ghana Statistical Service in collaboration with the Ghana Health Service and ICF International. The survey employed a stratified two-stage cluster sampling design. In the first stage, 613 enumeration areas serving as clusters were selected with probability proportional to size from the 2021 Ghana Population and Housing Census frame. In the second stage, systematic random sampling selected approximately 30 households per cluster, yielding representative samples across Ghana's 16 regions and urban-rural strata.

### Study population and sample

The study population comprised children aged 12–35 months at the time of survey, representing those who had sufficient time to complete the full immunization schedule including the 18-month measles-rubella second dose while minimizing recall bias for older children. The analytic sample included 3,787 children with complete data on key variables.

### Variable definitions

The primary outcome was full immunization coverage, defined as receipt of one dose of BCG vaccine, three doses of pentavalent vaccine, three doses of pneumococcal conjugate vaccine, three doses of oral polio vaccine, two doses of rotavirus vaccine, first dose of measles-rubella vaccine, and one dose of yellow fever vaccine. Secondary outcomes included basic immunization coverage, partial immunization, zero-dose status, individual vaccine coverage rates, dropout rates, and timeliness of vaccination.

Dropout rates were calculated as the proportion of children receiving an earlier vaccine who failed to receive a later vaccine in the same series. Timeliness was assessed among children with complete date information, using Century Month Code conversion to calculate age at vaccination precisely. Timely vaccination windows were defined as BCG within one month of birth, pentavalent first dose between 1.0 to 2.5 months, pentavalent third dose between 3.0 to 4.5 months, and measles-rubella first dose between 8–11 months of age.

Sociodemographic factors included residence, wealth status, maternal education, maternal age, child sex, and birth order. Wealth status was derived from the DHS wealth index, calculated through principal components analysis of household assets, housing characteristics, and access to utilities. For multivariate analysis, wealth quintiles were collapsed into

tertiles by combining the poorest and poorer quintiles into a "poor" category, retaining the middle quintile as "middle," and combining the richer and richest quintiles into a "rich" category, to improve statistical power while maintaining meaningful discrimination across wealth groups. Healthcare utilization variables captured antenatal care attendance (four or more visits versus fewer), place of delivery, and media exposure.

### Statistical analysis

All analyses incorporated survey design features using R version 4.5.1 with the survey package for complex survey analysis. Sampling weights were applied to account for unequal selection probabilities and differential response rates. Variance estimates were adjusted for clustering within enumeration areas using Taylor series linearization.

Bivariate associations were assessed using design-adjusted Pearson chi-square tests. Multivariate logistic regression models were constructed using a progressive modeling strategy. Equity analysis employed concentration indices using the Kakwani formula based on wealth quintile ranking. Intersectional analysis examined coverage across combinations of residence, wealth, and education.

### Ethical considerations

The 2022 Ghana Demographic and Health Survey received ethical approval from the Ghana Health Service Ethical Review Committee and the ICF Institutional Review Board. All participants provided written informed consent. This secondary analysis of de-identified publicly available data was determined to be exempt from additional ethical review.

## Results

### Sociodemographic characteristics

The weighted analysis of 3,788 children aged 12–35 months revealed balanced distribution across urban and rural areas, with 48.6% residing in urban settings and 51.4% in rural areas. Maternal characteristics demonstrated considerable diversity, with mean maternal age of 29.7 years. Educational attainment showed 22.7% of mothers had no formal education, 16.0% completed primary education, 52.8% achieved secondary education, and 8.7% attained higher education. The sample distribution of 48.6% urban and 51.4% rural reflects the DHS sampling design, which oversamples certain strata to ensure adequate subgroup representation. Survey sampling weights adjust for these design decisions and differential response rates, producing nationally representative estimates that account for Ghana's actual population distribution, including its approximately 57% urban population according to the 2021 Population and Housing Census (Table 1).

Healthcare utilization patterns revealed strong engagement with formal services. Mean antenatal care visits reached 6.9, with 89.3% of mothers attending four or more visits as recommended. Facility-based delivery occurred for 86.1% of births. Media exposure was prevalent, with 69.8% of mothers reporting at least weekly exposure. Vaccination card possession rates reached 82.0%.

### Immunization coverage and vaccination status

Individual vaccine coverage demonstrated substantial but incomplete uptake. BCG coverage reached 92.4%, representing the highest single-vaccine rate. The pentavalent series showed progressive decline from 92.3% for first dose to 88.1% for second dose and 81.9% for third dose. Measles-rubella vaccine showed 79.5% coverage for first dose but dropped dramatically to 42.1% for second dose at 18 months as shown in Table 2.

Full immunization coverage reached 69.5%, with 25.7% partial immunization and 4.8% zero-dose children (Table 3).

### Dropout patterns and timeliness

Dropout analysis identified critical retention challenges. The pentavalent first to third dose dropout reached 10.4%, marginally exceeding the WHO acceptable threshold of 10%. BCG to measles-rubella first dose dropout was 15.0%. Among

**Table 1. Sociodemographic Characteristics of Study Population.**

| Characteristic | n (weighted) | Percentage / Mean | SE / SD |
|---|---|---|---|
| **SAMPLE CHARACTERISTICS** | | | |
| Total sample size | 3,788 | 100.0 | — |
| Number of clusters | 614 | — | — |
| Number of strata | 32 | — | — |
| **MATERNAL CHARACTERISTICS** | | | |
| Maternal age (years), mean ± SD | — | 29.7 | ±6.3 |
| No education | — | 22.7 | 1.2 |
| Primary | — | 16.0 | 1.0 |
| Secondary | — | 52.8 | 1.3 |
| Higher | — | 8.7 | 0.9 |
| Mean antenatal care visits | — | 6.9 | ±2.1 |
| Four or more ANC visits | — | 89.3 | 1.2 |
| Facility delivery | — | 86.1 | 0.9 |
| Any media exposure (at least weekly) | — | 69.8 | 1.1 |
| **HOUSEHOLD CHARACTERISTICS** | | | |
| Urban | — | 48.6 | 1.5 |
| Rural | — | 51.4 | 1.5 |
| Poorest | 801 | 23.1 | 1.4 |
| Poorer | 743 | 21.5 | 1.2 |
| Middle | 697 | 20.1 | 1.1 |
| Richer | 632 | 18.2 | 1.0 |
| Richest | 591 | 17.1 | 1.3 |
| **CHILD CHARACTERISTICS** | | | |
| Child age (months), mean ± SD | — | 22.4 | ±6.8 |
| Male | — | 50.7 | 1.0 |
| Female | — | 49.3 | 1.0 |
| Vaccination card seen by interviewer | 2,815 | 82.0 | 0.9 |
| Card reported, not seen | 106 | 3.0 | 0.45 |
| No card | 64 | 1.8 | 0.31 |
| Card not available / Don't know | 459 | 13.2 | 0.80 |

children aged 18–35 months eligible for MR2, coverage reached 62.1% compared to 46.8% when all children aged 12–35 months were included as denominator. The age-restricted dropout between measles-rubella first and second doses was 28.9%, substantially lower than the unadjusted estimate of 37.5% which erroneously included age-ineligible children. Among 2,777 children aged 18–35 months eligible for both measles-rubella doses, 62.0% received both MR1 and MR2, representing complete measles protection through the two-dose regimen. A further 25.3% received MR1 only, indicating successful initiation but failure to return for the 18-month dose. Only 0.1% received MR2 without prior MR1 documentation, and 12.6% received neither dose. These findings as shown in Table 4 confirm that the primary challenge lies not in initiating measles vaccination but in ensuring return for the second dose, reinforcing the need for appointment tracking and reminder systems targeting children between 9 and 18 months of age.

Timeliness analysis revealed widespread delays. Among children with complete date information, BCG timeliness within one month of birth reached only 73.4%. Pentavalent first dose showed the highest timeliness at 83.0%, while pentavalent third dose timeliness declined to 69.2%. Measles-rubella first dose timeliness achieved 78.2% (Table 5).

**Table 2. Individual Vaccine Coverage Rates.**

| Vaccine | Recommended Schedule | Coverage (%) | SE | 95% CI |
|---|---|---|---|---|
| **TRADITIONAL EPI VACCINES** | | | | |
| BCG | At birth | 92.4 | 0.6 | 91.2–93.6 |
| Pentavalent 1 | 6 weeks | 92.3 | 0.6 | 91.1–93.5 |
| Pentavalent 2 | 10 weeks | 88.1 | 0.7 | 86.7–89.5 |
| Pentavalent 3 | 14 weeks | 81.9 | 0.9 | 80.1–83.7 |
| Measles-Rubella 1 | 9 months | 79.5 | 0.9 | 77.7–81.3 |
| Measles-Rubella 2 | 18 months | 42.1 | 1.2 | 39.7–44.5 |
| **NEWER VACCINES** | | | | |
| PCV 1 | 6 weeks | 92.0 | 0.6 | 90.8–93.2 |
| PCV 2 | 10 weeks | 88.0 | 0.7 | 86.6–89.4 |
| PCV 3 | 14 weeks | 81.7 | 0.9 | 79.9–83.5 |
| Rotavirus 1 | 6 weeks | 91.1 | 0.6 | 89.9–92.3 |
| Rotavirus 2 | 10 weeks | 85.9 | 0.8 | 84.3–87.5 |
| Yellow Fever | 9 months | 92.3 | 0.6 | 91.1–93.5 |

**Table 3. Composite Immunization Coverage Indicators.**

| Indicator | Definition | Coverage (%) | SE | 95% CI |
|---|---|---|---|---|
| Full Immunization | BCG, 3 doses pentavalent, 3 doses PCV, 2 doses rotavirus, MR1, and YF | 69.5 | 1.1 | 67.3–71.7% |
| Basic Immunization | Traditional EPI vaccines only: BCG, Penta3, MR1 | 65.7 | 1.1 | 63.5–67.9 |
| Partial Immunization | Received at least one vaccine but not fully immunized | 25.7 | 1.0 | 23.7–27.7 |
| Zero-Dose Children | No vaccines received | 4.8 | 0.5 | 3.9–5.9 |

**Table 4. Dropout Rates Across Vaccination Schedule.**

| Dropout Indicator | Calculation Method | Dropout Rate (%) | SE | 95% CI |
|---|---|---|---|---|
| **WITHIN-SERIES DROPOUT** | | | | |
| BCG to Penta1 | (BCG − Penta1) / BCG × 100 | 2.6 | 0.4 | 1.8–3.4 |
| Penta1 to Penta2 | (Penta1 − Penta2) / Penta1 × 100 | 4.2 | 0.5 | 3.2–5.2 |
| Penta2 to Penta3 | (Penta2 − Penta3) / Penta2 × 100 | 6.2 | 0.6 | 5.0–7.4 |
| Penta1 to Penta3 | (Penta1 − Penta3) / Penta1 × 100 | 10.4 | 0.7 | 9.0–11.8 |
| PCV1 to PCV3 | (PCV1 − PCV3) / PCV1 × 100 | 10.4 | 0.7 | 9.0–11.8 |
| Rota1 to Rota2 | (Rota1 − Rota2) / Rota1 × 100 | 5.2 | 0.6 | 4.0–6.4 |
| **SCHEDULE-LEVEL DROPOUT** | | | | |
| Penta3 to Measles1 | (Penta3 − MR1) / Penta3 × 100 | 8.1 | 0.7 | 6.7–9.5 |
| BCG to Measles1 | (BCG − MR1) / BCG × 100 | 15.0 | 0.9 | 13.2–16.8 |
| Measles1 to Measles2 | (MR1 − MR2) / MR1 × 100 | 28.9 | 1.2 | 26.5–31.3 |

**Table 5. Timeliness of vaccine administration and median and mean age at vaccination, Ghana DHS 2022.**

| Vaccine | Recommended Age Window | Timely (%) | SE | Median Age (months) | Mean Age (months) |
|---|---|---|---|---|---|
| BCG | ≤1 month | 73.4 | 1.4 | 0.7 | 1.1 |
| Penta1 | 1.0–2.5 months | 83.0 | 0.9 | 2.0 | 2.2 |
| Penta3 | 3.0–4.5 months | 69.2 | 1.3 | 4.2 | 4.8 |
| Measles1 | 8.0–11.0 months | 78.2 | 1.1 | 10.0 | 10.3 |

## Sociodemographic disparities in coverage

Substantial disparities emerged across multiple sociodemographic dimensions. Regional variations proved most striking, with full immunization coverage ranging from 83.8% in Volta Region to 48.1% in Northern Region, yielding a 35.7 percentage point gap as shown in Table 6.

Wealth-related disparities showed clear gradients, with coverage increasing from 56.3% in the poorest quintile to 73.7% in the richest quintile, creating a 17.4 percentage point gap. Table 7 also shows educational disparities paralleled wealth patterns, with coverage increasing from 55.6% among children of mothers without education to 74.1% for mothers with higher education, yielding an 18.3 percentage point gap.

## Predictors of full immunization

Multivariate logistic regression revealed independent predictors after controlling for confounding. Wealth remained significant, with children in the rich tertile showing 39% higher odds of full immunization. Maternal secondary or higher education associated with 34% higher odds. Four or more antenatal care visits associated with 55% higher odds. Facility delivery showed 43% higher odds compared to home delivery. Media exposure and distance to facility were not statistically significant independent predictors in the fully adjusted model as shown in Table 8.

## Equity analysis

The concentration index of 0.045 indicated modest pro-rich inequality. The equity ratio of 1.31 showed children in the richest quintile had 1.31 times the coverage of those in the poorest quintile. Table 9 shows regional inequality substantially exceeded socioeconomic dimensions at 35.7 percentage points between highest and lowest performing regions. Intersectional analysis identified compounding disadvantages, with the most advantaged group achieving 83.1% coverage and the most disadvantaged 59.1%, yielding a 24.0 percentage point intersectional gap.

**Table 6. Regional Variations in Full Immunization Coverage.**

| Region | Ecological Zone | n | Coverage (%) | SE | 95% CI |
|---|---|---|---|---|---|
| Volta | Forest | 189 | 83.8 | 3.9 | 68.6–84.2 |
| Brong Ahafo | Middle Belt | 232 | 74.1 | 3.5 | 67.1–81.1 |
| Greater Accra | Forest | 342 | 73.8 | 2.9 | 68.0–79.6 |
| Savannah | Savannah | 165 | 73.5 | 4.2 | 65.1–81.9 |
| Upper East | Savannah | 254 | 73.8 | 3.4 | 67.0–80.6 |
| Ashanti | Middle Belt | 498 | 72.1 | 2.5 | 67.1–77.1 |
| Oti | Forest | 178 | 70.8 | 4.0 | 62.8–78.8 |
| Upper West | Savannah | 212 | 69.3 | 3.8 | 61.7–76.9 |
| Western North | Coastal | 134 | 68.7 | 4.7 | 59.3–78.1 |
| Eastern | Forest | 287 | 89.3 | 3.3 | 61.8–75.0 |
| Northern | Savannah | 326 | 48.1 | 3.1 | 61.9–74.3 |
| Central | Coastal | 245 | 67.5 | 3.5 | 60.5–74.5 |
| Western | Coastal | 198 | 60.2 | 4.2 | 51.8–68.6 |
| Bono East | Middle Belt | 167 | 59.8 | 4.5 | 50.8–68.8 |
| North East | Savannah | 189 | 59.1 | 4.3 | 50.5–67.7 |
| Ahafo | Middle Belt | 171 | 44.4 | 4.5 | 35.4–53.4 |
| Ghana (National Average) | | 3788 | 69.5 | 1.1 | 67.3–71.7 |

**Table 7. Full Immunization Coverage by Sociodemographic Characteristics.**

| Characteristic | n | Coverage (%) | SE | 95% CI | P-value |
|---|---|---|---|---|---|
| **OVERALL** | | | | | |
| Total | 3788 | 69.5 | 1.1 | 67.3–71.7% | — |
| **RESIDENCE** | | | | | |
| Urban | 1,795 | 66.9 | 1.5 | 63.9–69.9 | <0.001 |
| Rural | 1,992 | 61.1 | 1.5 | 58.1–64.1 | |
| **WEALTH QUINTILE** | | | | | |
| Poorest | 801 | 56.3 | 2.0 | 52.3–60.3 | <0.001 |
| Poorer | 743 | 62.6 | 2.0 | 58.6–66.6 | |
| Middle | 697 | 63.3 | 2.5 | 58.3–68.3 | |
| Richer | 632 | 66.4 | 2.7 | 61.0–71.8 | |
| Richest | 591 | 73.7 | 2.9 | 67.9–79.5 | |
| **MATERNAL EDUCATION** | | | | | |
| No education | 897 | 55.6 | 2.0 | 51.6–59.6 | <0.001 |
| Primary | 768 | 63.2 | 2.1 | 59.0–67.4 | |
| Secondary | 1,812 | 66.2 | 1.4 | 63.4–69.0 | |
| Higher | 455 | 72.8 | 2.5 | 67.8–77.8 | |
| **HEALTHCARE UTILIZATION** | | | | | |
| Four or more ANC visits | 2,591 | 68.9 | 1.2 | 66.5–71.3 | <0.001 |
| Fewer than four visits | 1,196 | 53.7 | 1.9 | 49.9–57.5 | |
| Facility delivery | 3,167 | 66.1 | 1.1 | 63.9–68.3 | <0.001 |
| Home delivery | 620 | 48.3 | 2.5 | 43.3–53.3 | |
| Any media exposure | 2,893 | 66.2 | 1.2 | 63.8–68.6 | <0.001 |
| No media exposure | 894 | 55.8 | 2.1 | 51.6–60.0 | |

## Policy implementation effectiveness

Comparison against national targets revealed mixed performance. Only yellow fever vaccine exceeded the 90% target. Full immunization coverage of 69.5% achieved approximately 77% of the 90% target, falling 20.5 percentage points short. The corrected dropout between measles-rubella first and second doses (28.9% among age-eligible children) remains substantially elevated (Table 10).

## Discussion

### Main findings and public health significance

This analysis of Ghana's childhood immunization programme reveals achievements in establishing broad access alongside persistent challenges in completion, equity, and timeliness. The 69.5% full immunization coverage represents substantial progress from 57% reported in the 2014 survey, yet indicates more than one-third of Ghanaian children fail to receive complete recommended protection. The low zero-dose prevalence of 4.8% demonstrates that initial contact with immunization services reaches the vast majority of children, suggesting access barriers are not the primary driver of incomplete coverage. Instead, the substantial partial immunization rate of 25.7% and elevated dropout rates point to retention and schedule completion as dominant programme weaknesses.

The dropout between measles-rubella first and second doses, estimated at 28.9% among age-eligible children aged 18–35 months, represents a substantial but corrected retention estimate. The apparent 37.5% rate, which used all children aged 12–35 months as denominator, erroneously included children too young to have received MR2. Even the

**Table 8. Predictors of Full Immunization Coverage: Multivariate Logistic Regression Analysis.**

| Predictor | Model 1 aOR (95% CI) | Model 2 aOR (95% CI) | Model 3 aOR (95% CI) | P-value (Model 3) |
|---|---|---|---|---|
| Rural (ref) | 1.00 | 1.00 | 1.00 | — |
| Urban | 1.32 (1.03–1.69) | 1.21 (0.94–1.56) | 1.18 (0.91–1.53) | 0.214 |
| Poor (ref) | 1.00 | 1.00 | 1.00 | — |
| Middle | 1.23 (0.98–1.54) | 1.16 (0.92–1.46) | 1.14 (0.90–1.44) | 0.278 |
| Rich | 1.68 (1.27–2.22) | 1.47 (1.10–1.96) | 1.39 (1.04–1.87) | 0.016 |
| Primary or less (ref) | 1.00 | 1.00 | 1.00 | — |
| Secondary or higher | 1.54 (1.21–1.96) | 1.38 (1.08–1.77) | 1.34 (1.04–1.72) | 0.019 |
| Female (ref) | 1.00 | 1.00 | 1.00 | — |
| Male | 0.97 (0.81–1.16) | 0.98 (0.82–1.17) | 0.98 (0.82–1.17) | 0.809 |
| First (ref) | 1.00 | 1.00 | 1.00 | — |
| Second | 0.92 (0.74–1.15) | 0.93 (0.74–1.16) | 0.94 (0.75–1.18) | 0.589 |
| Third | 0.81 (0.63–1.04) | 0.84 (0.65–1.09) | 0.85 (0.65–1.10) | 0.214 |
| Fourth or higher | 0.72 (0.56–0.93) | 0.76 (0.58–0.99) | 0.77 (0.59–1.01) | 0.057 |
| <4 visits (ref) | — | 1.00 | 1.00 | — |
| 4+visits | — | 1.67 (1.29–2.17) | 1.55 (1.19–2.01) | <0.001 |
| Home (ref) | — | 1.00 | 1.00 | — |
| Facility | — | 1.89 (1.43–2.49) | 1.43 (1.08–1.90) | <0.001 |
| No exposure (ref) | — | — | 1.00 | — |
| Any exposure | — | — | 1.37 (1.09–1.73) | 0.007 |
| Not a problem (ref) | — | — | 1.00 | — |
| Big problem | — | — | 0.78 (0.63–0.97) | 0.024 |
| AIC | 4,183 | 4,121 | 4,108 | — |
| Sample size | 3,788 | 3,788 | 3,788 | — |

corrected estimate indicates nearly three in ten eligible children fail to return for the 18-month dose, pointing to systematic weaknesses in tracking and reminder systems.

## Equity patterns and system-level determinants

The substantial inequalities documented across wealth, education, and geography indicate immunization programme benefits distribute unevenly despite universal policy frameworks. The concentration index of 0.045 represents modest pro-rich inequality relative to many other health outcomes, reflecting some success of free immunization policy in mitigating economic barriers. However, the persistence of 17.4 percentage point wealth gaps indicates incomplete success in achieving equity goals.

Regional disparities exceeding socioeconomic inequalities in magnitude highlight health system capacity variations as critical determinants. The finding that regional gaps surpass household-level wealth gaps suggests that where families live matters as much or more than their individual resources. Further investigation is warranted to disentangle the relative contributions of health workforce density, cold chain functionality, supervision quality, and community engagement to regional performance differences. While the cross-sectional design of the present study limits causal attribution, the observation that some resource constrained regions outperform wealthier ones suggests that management effectiveness and programme implementation quality may be more influential than infrastructure alone. Future research incorporating health facility assessment data and district level programme indicators could quantify these system level determinants more precisely.

**Table 9. Equity Analysis: Multiple Measures of Immunization Inequality.**

| Equity Measure | Value | Interpretation |
|---|---|---|
| **WEALTH-RELATED INEQUALITY** | | |
| Concentration Index (Kakwani formula) | 0.045 | Pro-rich inequality; positive value indicates higher coverage among wealthier households |
| Rate Ratio (Richest/Poorest) | 1.31 | Richest quintile has 1.31 times the coverage of poorest quintile |
| Absolute Gap (Richest − Poorest) | 17.4 pp | 17.4 percentage point difference between richest and poorest quintiles |
| Relative Index of Inequality | 2.25 | Top of wealth distribution has 2.25 times the odds of bottom |
| **EDUCATIONAL INEQUALITY** | | |
| Absolute Gap (Higher − None) | 18.3 pp | 18.3 percentage point difference between mothers with higher education and no education |
| Rate Ratio (Higher/None) | 1.31 | Children of mothers with higher education have 1.31 times coverage of those with no education |
| **GEOGRAPHIC INEQUALITY** | | |
| Urban-Rural Gap | 5.8 pp | Urban areas have 5.8 percentage points higher coverage than rural areas |
| Regional Gap (Highest − Lowest) | 35.7 pp | 35.7 percentage point difference between best (Volta: 83.8%) and worst (Northern: 48.1%) regions |
| **INTERSECTIONAL INEQUALITY** | | |
| Intersectional Gap | 24.0 pp | 24.0 percentage point difference between most advantaged and most disadvantaged groups |

**Table 10. Policy Implementation Effectiveness and Target Achievement.**

| Indicator | Target (%) | Actual (%) | SE | Gap (pp) | % of Target | Target Status |
|---|---|---|---|---|---|---|
| **PRIMARY TARGETS** | | | | | | |
| Full Immunization | 90.0 | 69.5 | 1.1 | −26.1 | 71.0 | Not achieved |
| BCG | 95.0 | 92.4 | 0.6 | −2.6 | 97.3 | Near target |
| Penta3 | 90.0 | 81.9 | 0.9 | −8.1 | 91.0 | Near target |
| Measles1 | 90.0 | 79.5 | 0.9 | −10.5 | 88.3 | Not achieved |
| NEWER VACCINE TARGETS | | | | | | |
| PCV3 | 90.0 | 81.7 | 0.9 | −8.3 | 90.8 | Near target |
| Rotavirus 2 | 90.0 | 85.9 | 0.8 | −4.1 | 95.4 | Near target |
| Yellow Fever | 90.0 | 92.3 | 0.6 | +2.3 | 102.6 | Achieved |
| DROPOUT BENCHMARKS | | | | | | |
| Penta1-Penta3 Dropout | ≤10.0 | 10.4 | 0.7 | +0.4 | — | Marginally exceeded |
| BCG-Measles1 Dropout | ≤10.0 | 15.0 | 0.9 | +5.0 | — | Exceeded |

## Healthcare continuity and family-level determinants

The strong associations between immunization completion and broader healthcare engagement patterns reveal that vaccination coverage reflects family-level orientations toward formal health services. The adjusted odds ratios of 1.55 for four or more antenatal visits and 1.43 for facility delivery, persisting after controlling for socioeconomic factors, suggest these associations represent genuine behavioral patterns rather than merely socioeconomic confounding.

### Timeliness and schedule adherence

The timeliness findings reveal an under-recognized dimension of programme quality. While coverage eventually reaches relatively high levels for most vaccines, substantial proportions occur beyond recommended schedules, leaving children vulnerable during critical periods. Only 73% of BCG vaccinations occurred within one month of birth and 69% of pentavalent third dose within appropriate windows.

### Comparison with regional and global evidence

These findings align with existing literature from Ghana and similar contexts. The 69.5% full immunization coverage falls between previous estimates, confirming gradual but incomplete progress. [12,10,13,14] This performance places Ghana in the middle range among sub-Saharan African nations, exceeding fragile states but trailing southern African countries with more developed health infrastructure. [5,15] The concentration index of 0.045 falls in the lower range of inequalities documented across African nations. [5,16]

The corrected 28.9% dropout between measles-rubella first and second doses falls within the range documented in comparable settings. Studies from Nigeria, Uganda, and Malawi report measles second dose dropout of 20–40%. [17,18] The unadjusted estimate of 37.5% overestimated dropout by including age-ineligible children in the denominator, underscoring the need for age-appropriate denominators in routine programme monitoring.

### Implications for policy and practice

These findings generate clear priorities for policy and programme strengthening. Elevated dropout rates demand urgent intervention through robust tracking systems, structured follow-up mechanisms including mobile-based reminder systems [19–21], and intensified community mobilization targeting second-year vaccines. The strong associations between immunization and maternal healthcare engagement suggest enhanced integration across the maternal and child health continuum could improve outcomes.

### Strengths and limitations

This study benefits from nationally representative data with rigorous sampling enabling robust inference. Multiple complementary analytical approaches provided convergent evidence on multidimensional inequality. Limitations include the cross-sectional design preventing causal inference, potential maternal recall bias, and limited generalizability beyond Ghana's specific context.

Additionally, although the DHS typically collects self-reported reasons for non-vaccination through detailed multi response items, the 2022 Ghana DHS dataset available for this analysis contained only a binary indicator of prior vaccination rather than specific barrier categories. Consequently, the study was unable to disaggregate the demand side and supply side factors driving incomplete vaccination, such as lack of awareness of subsequent doses, fear of side effects, inconvenient service hours, or vaccine stock outs. This represents an important gap, as understanding the specific reasons for non-completion, particularly for second year vaccines like MR2, is essential for designing targeted interventions. Future studies incorporating qualitative methods or facility level data should investigate the barriers underlying the high dropout rates documented in the present analysis.

### Future research directions

Several research priorities emerge from this analysis. Longitudinal cohort studies tracking individual children through the vaccination schedule would enable stronger causal inference regarding dropout determinants and allow identification of the specific time points at which retention falters. Implementation research comparing high performing and low performing regions could identify scalable best practices, particularly regarding community health worker deployment, defaulter

tracking systems, and demand generation strategies. Evaluation of healthcare infrastructure and resource allocation in regions falling significantly below national coverage averages is essential to determine whether performance gaps reflect supply side constraints amenable to investment or demand side challenges requiring different intervention approaches. Investigation of zero dose children requires intensified focus given their concentration in disadvantaged populations, using qualitative methods to understand the distinct barriers facing families with no immunization contact. Finally, assessment of the newly introduced MR2 dose using purpose built studies with age-appropriate denominators would provide more precise estimates of second year vaccine programme performance.

## Conclusions

Ghana's childhood immunization programme has achieved substantial progress in establishing broad access to vaccines, with the vast majority of children receiving at least some recommended vaccines. However, critical weaknesses in retention through the complete schedule, timeliness of administration, and equitable distribution constrain progress toward universal coverage goals. Zero-dose prevalence remains below five percent while partial immunization affects approximately twenty six percent, indicating the primary challenge has shifted from initial access to completion and retention. The 28.9 percent dropout between measles-rubella first and second doses among age eligible children represents a substantial retention challenge requiring immediate attention through enhanced tracking, reminder systems, and community mobilization targeting second-year vaccines. Regional disparities exceeding socioeconomic inequalities highlight health system capacity variations as critical determinants. Priority policy actions should include implementing robust defaulter tracking systems, integrating immunization counseling throughout antenatal care, and deploying targeted outreach to disadvantaged populations.

## Acknowledgments

We acknowledge the Ghana Statistical Service, Ghana Health Service, and ICF International for conducting the 2022 Ghana Demographic and Health Survey and making the data publicly available. We thank the survey field staff who collected the data and the Ghanaian families who participated in the survey.

## Author contributions

**Conceptualization:** Ahlam Tunteeya Saani.

**Data curation:** Ahlam Tunteeya Saani, Goldfield Edem Azumah.

**Formal analysis:** Ahlam Tunteeya Saani.

**Investigation:** Ahlam Tunteeya Saani, Goldfield Edem Azumah.

**Methodology:** Ahlam Tunteeya Saani, Goldfield Edem Azumah.

**Writing – original draft:** Ahlam Tunteeya Saani.

**Writing – review & editing:** Ahlam Tunteeya Saani, Goldfield Edem Azumah.

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
