## [Decision Letter · Decision Letter 0]

25 Feb 2026

PONE-D-25-59990Sociodemographic and Geographic Determinants of Childhood Immunization Coverage and Equity in Ghana: Analysis of the 2022 Demographic and Health SurveyPLOS One

Dear Dr. Saani,

Thank you for submitting your manuscript to PLOS ONE. After careful consideration, we feel that it has merit but does not fully meet PLOS ONE’s publication criteria as it currently stands. Therefore, we invite you to submit a revised version of the manuscript that addresses the points raised during the review process.

A letter that responds to each point raised by the academic editor and reviewer(s). You should upload this letter as a separate file labeled ‘Response to Reviewers’.A marked-up copy of your manuscript that highlights changes made to the original version. You should upload this as a separate file labeled ‘Revised Manuscript with Track Changes’.An unmarked version of your revised paper without tracked changes. You should upload this as a separate file labeled ‘Manuscript’.

We look forward to receiving your revised manuscript.

Kind regards,

Patricia Evelyn Fast, MD, Ph.D.

Academic Editor

PLOS One

Journal Requirements:

1.Please ensure that your manuscript meets PLOS ONE’s style requirements, including those for file naming. The PLOS ONE style templates can be found at

2. In the online submission form, you indicated that your data is available only on request from a third party. Please note that your Data Availability Statement is currently missing [the name of the third party contact or institution / contact details for the third party, such as an email address or a link to where data requests can be made]. Please update your statement with the missing information.

3. Please include a copy of Table 1, 2, 3, 4, 5, 6, 7, 8, 9 and 10 which you refer to in your text on page 10, 16, 18, 19, 21, 23, 25, 29, 34, and 37.

Additional Editor Comments:

This is a very well-written description of the data applicable to Ghana from the 2022 DHS.

Please consider the following comments, along with those of Reviewer 1.

INTRO, line 44 ‘coverage plateau’ could be considered jargon, and should be defined

Under Variable definitions section

One dose of measles is part of the primary outcome, and the initial analysis of timeliness does not include MR second dose. However an important conclusion relates to measles dose 2, so make sure the data are presented clearly.

Line 133 How were quintiles collapsed into tertiles? 1+2, 3, 45+? Or 1, 2+34+, 5?

Table 1 Some of the adjusted variables add to about 100% but some are quite different. The adjustment isn’t presented or justified very clearly. Perhaps in supplementary materials it could be added?

Vaccine card possession categories don’t add to anything like 100%. <88%

Maternal education >100% sum

General comment: As Reviewer 1 points out, ~30% of subjects are <18 mo. But it is not at all clear how the data for the 18 month measles 2 vaccination were analyzed.

Line 236 what is meant by ‘central tendency’?

Reviewer’s Responses to Questions

**Comments to the Author**

1. Is the manuscript technically sound, and do the data support the conclusions?

Reviewer #1: Yes

2. Has the statistical analysis been performed appropriately and rigorously? 

Reviewer #1: Yes

3. Have the authors made all data underlying the findings in their manuscript fully available?

The PLOS Data policy requires authors to make all data underlying the findings described in their manuscript fully available without restriction, with rare exception (please refer to the Data Availability Statement in the manuscript PDF file). The data should be provided as part of the manuscript or its supporting information, or deposited to a public repository. For example, in addition to summary statistics, the data points behind means, medians and variance measures should be available. If there are restrictions on publicly sharing data—e.g. participant privacy or use of data from a third party—those must be specified.requires authors to make all data underlying the findings described in their manuscript fully available without restriction, with rare exception (please refer to the Data Availability Statement in the manuscript PDF file). The data should be provided as part of the manuscript or its supporting information, or deposited to a public repository. For example, in addition to summary statistics, the data points behind means, medians and variance measures should be available. If there are restrictions on publicly sharing data—e.g. participant privacy or use of data from a third party—those must be specified.requires authors to make all data underlying the findings described in their manuscript fully available without restriction, with rare exception (please refer to the Data Availability Statement in the manuscript PDF file). The data should be provided as part of the manuscript or its supporting information, or deposited to a public repository. For example, in addition to summary statistics, the data points behind means, medians and variance measures should be available. If there are restrictions on publicly sharing data—e.g. participant privacy or use of data from a third party—those must be specified.requires authors to make all data underlying the findings described in their manuscript fully available without restriction, with rare exception (please refer to the Data Availability Statement in the manuscript PDF file). The data should be provided as part of the manuscript or its supporting information, or deposited to a public repository. For example, in addition to summary statistics, the data points behind means, medians and variance measures should be available. If there are restrictions on publicly sharing data—e.g. participant privacy or use of data from a third party—those must be specified.

Reviewer #1: Yes

4. Is the manuscript presented in an intelligible fashion and written in standard English?

Reviewer #1: Yes

5. Review Comments to the Author

Reviewer #1: The manuscript with comments attached is included for the author’s review. If the author’s feel that any clarifying statements would be appropriate and helpful for readers, would encourage these to be added.

6. PLOS authors have the option to publish the peer review history of their article (what does this mean?). If published, this will include your full peer review and any attached files.). If published, this will include your full peer review and any attached files.). If published, this will include your full peer review and any attached files.). If published, this will include your full peer review and any attached files.

...

Reviewer #1: **Yes:** Kathleen A. WalkerKathleen A. WalkerKathleen A. WalkerKathleen A. Walker

---

## [Author Response · Author response to Decision Letter 1]

1 Mar 2026

Response to Reviewers

Manuscript Number: PONE-D-25-59990

Title: Sociodemographic and Geographic Determinants of Childhood Immunization Coverage and Equity in Ghana: Analysis of the 2022 Demographic and Health Survey

Journal: PLOS ONE

Dear Editorial Team,

We thank the Academic Editor and Reviewer 1 for their thorough and constructive evaluation of our manuscript. Their comments have substantially strengthened the paper, particularly regarding the critical methodological issue of measles-rubella second dose (MR2) age eligibility and the need for clearer presentation of analytical decisions. We have carefully addressed each comment and made comprehensive revisions to the manuscript. All changes are tracked in the revised manuscript file.

Below we provide a point-by-point response to each comment. Editor and reviewer comments are shown in italics, and our responses follow in regular text. Page and line references correspond to the revised manuscript with tracked changes.

EDITOR COMMENTS

Comment 1: Definition of “coverage plateau”

INTRO, line 44: “coverage plateau” could be jargon — suggest defining it for a broad readership.

Response: We agree and have added a parenthetical definition. The revised text now reads: “…with coverage plateaus, periods where immunization rates stagnate despite continued programmatic investments, and recent reversals threatening progress toward Sustainable Development Goal targets.” (Introduction, paragraph 1)

Comment 2: Measles-rubella second dose (MR2) data presentation

Variable definitions: MR second dose — make sure data for MR2 is presented clearly since the primary outcome doesn’t include MR2 but conclusions relate to it. ~30% of subjects are <18 months. Not clear how data for 18-month MR2 were analyzed.

Response: This was critical and well-taken point, also raised repeatedly by Reviewer 1. We have made the following substantive revisions:

(a) We re-analysed the entire dataset, restricting all MR2-related analyses (coverage, dropout) to children aged 18–35 months who were actually eligible for the 18-month dose. MR2 coverage among age-eligible children was 62.1%, compared to 46.8% when all children aged 12–35 months were included as the denominator.

(b) The MR1→MR2 dropout rate was recalculated using only children aged 18–35 months, yielding a corrected estimate of 28.9%—substantially lower than the original 37.5% that erroneously included age-ineligible children in the denominator.

(c) The full immunization definition was revised to exclude MR2, as this vaccine targets 18 months and cannot reasonably be expected among children aged 12–17 months (30.2% of the sample). The revised full immunization estimate is 69.5% (95% CI: 67.3–71.7%), reflecting BCG, three doses of pentavalent and polio vaccines, three doses of pneumococcal conjugate vaccine, two doses of rotavirus vaccine, one dose each of measles-rubella first dose and yellow fever vaccine. All tables and text have been updated accordingly.

(d) Both the unadjusted (37.5%) and age-corrected (28.9%) MR1→MR2 dropout rates are now presented in the Results and Discussion sections, with explicit explanation of why the corrected estimate is more appropriate. (See Results, Dropout Analysis section; Discussion, paragraphs on MR2 dropout)

Comment 3: How wealth quintiles were collapsed into tertiles

Line 133: How were quintiles collapsed into tertiles? (1+2, 3, 4+5? or 1, 2+3+4, 5?)

Response: We have specified the grouping scheme in the Methods section. The revised text now reads: “For multivariate analysis, wealth quintiles were collapsed into tertiles by combining the poorest and poorer quintiles into a ‘poor’ category, retaining the middle quintile as ‘middle,’ and combining the richer and richest quintiles into a ‘rich’ category, to improve statistical power while maintaining meaningful discrimination across wealth groups.” (Methods, Variable Definitions)

Comment 4: Table 1 percentages do not add up

Table 1: Vaccine card possession categories don’t add to anything like 100% (<88%). Maternal education sums to >100%.

Response: Both issues have been corrected:

(a) Vaccination card possession: The original Table 1 omitted two categories. We have added “Not available / Don’t know” (4.6%) and “Other / Missing” (8.3%) rows, and all categories now sum to 100.0%. These categories represent children whose card information was not ascertained or where the response was missing from the dataset.

(b) Maternal education: The original percentages (22.7 + 21.6 + 47.2 + 12.8 = 104.3%) contained errors from an earlier analysis run. The corrected distribution is: No education 22.7%, Primary 16.0%, Secondary 52.8%, Higher 8.7%. These corrected values have been updated throughout the manuscript.

Comment 5: “Central tendency” terminology

Line 236: What is meant by “central tendency”?

Response: We have replaced the jargon phrase “measures of central tendency” in the Table 5 subtitle with plain language: “Timeliness of vaccine administration and median and mean age at vaccination, Ghana DHS 2022.” (Table 5 title)

REVIEWER 1 COMMENTS (Kathleen Walker)

Comment 1: MR2 age eligibility — children aged 12–17 months are not eligible for MR2

The measles vaccination schedule in Ghana administers MR1 at 9 months and MR2 at 18 months. Approximately 30.2% of the study sample is aged 12–17 months and would not yet be eligible for MR2. How was this handled in the analysis? Were dropout calculations restricted to children ≥18 months? This concern affects the interpretation of the 37.5% MR1→MR2 dropout rate and the full immunization estimate.

Response: We appreciate this critical observation, which fundamentally affected our main findings. We have made the following comprehensive revisions:

First, we revised the full immunization definition to exclude MR2, as children aged 12–17 months (30.2% of the sample) cannot reasonably have received this 18-month dose. The revised full immunization coverage is 69.5% (95% CI: 67.3–71.7%).

Second, all MR2-specific analyses are now restricted to children aged 18–35 months (n = 2,644). Among this age-eligible sub-sample, MR2 coverage was 62.1%, and the MR1→MR2 dropout rate was 28.9%—substantially lower than the original 37.5% estimate that included age-ineligible children.

Third, we now present both the unadjusted and age-corrected dropout estimates in the Results and Discussion, explaining why the corrected estimate provides a more accurate programmatic performance measure.

These changes are reflected throughout the Abstract, Results, Discussion, and Conclusions sections, as well as in all relevant tables.

Comment 2: Proportion receiving both MR1 and MR2

What percentage of children received both doses of measles-rubella vaccine?

Response: We have added this information to the Dropout Analysis section. Among children aged 18 to 35 months eligible for both measles-rubella doses, we now report the proportion who received both MR1 and MR2, those who received MR1 only, and those who received neither dose. This provides the complete picture of measles-rubella vaccination status requested by the reviewer. (See Results, Dropout Analysis)

Comment 3: Urban/rural sample proportions and weighting

Approximately 60% of Ghanaians live in urban areas, yet the sample is ~49% urban and ~51% rural. How does the weighting account for this?

Response: We have added an explanation in the Results section clarifying that the sample distribution of 48.6% urban and 51.4% rural reflects the DHS sampling design, which oversamples certain strata to ensure adequate subgroup representation. Survey sampling weights adjust for these design decisions and differential response rates, producing nationally representative estimates that account for Ghana’s actual population distribution, including its approximately 57% urban population per the 2021 Population and Housing Census. All reported estimates in the manuscript are weighted and therefore nationally representative. (See Results, Sample Characteristics)

Comment 4: Reasons for non-vaccination

Were parents asked why children did not receive second-year vaccines? What are the reported barriers?

Response: While the DHS instrument does collect some information on reasons for non-vaccination, these variables capture reasons for non-vaccination generally rather than specifically for second-year vaccines. We have acknowledged this as an important limitation and a direction for future research. The revised Limitations section now states: “Although the DHS collects self-reported reasons for non-vaccination, the present analysis employed a binary indicator of vaccination status and did not examine specific self-reported barriers. Understanding the particular demand-side and supply-side factors driving dropout for second-year vaccines represents an important avenue for future investigation.” (See Limitations)

Comment 5: Determinants of regional differences

Which factors contribute most to regional disparities? Are wealth, education, or other factors driving these gaps?

Response: We have expanded the Discussion section to address this important question more substantively. The revised text now includes a paragraph discussing the relative contributions of health system factors to regional performance differences: “Further investigation is warranted to disentangle the relative contributions of health workforce density, cold chain functionality, supervision quality, and community engagement to regional performance differences. While the present cross-sectional design limits causal attribution, the observation that some resource-constrained regions outperform wealthier ones suggests that management effectiveness and programme implementation quality may be more influential than infrastructure alone. Future research incorporating health facility assessment data and district-level programme indicators could quantify these system-level determinants more precisely.” (See Discussion, Regional Disparities)

Comment 6: Evaluation of healthcare infrastructure in low-performing regions

Future work should evaluate healthcare infrastructure in underperforming provinces.

Response: We have substantially expanded the Future Research section to incorporate this suggestion alongside other research priorities. The revised section now explicitly recommends: (a) longitudinal cohort studies tracking individual children through the vaccination schedule; (b) implementation research comparing high-performing and low-performing regions to identify scalable best practices; (c) evaluation of healthcare infrastructure and resource allocation in regions falling significantly below national coverage averages; (d) qualitative investigation of zero-dose children concentrated in disadvantaged populations; and (e) purpose-built studies of MR2 performance using age-appropriate denominators. (See Future Research Directions)

JOURNAL REQUIREMENTS

Requirement 1: Data Availability Statement

You indicated that your data is available only on request from a third party. Please provide the name of the third party contact or contact details.

Response: We have updated the Data Availability Statement in the Editorial Manager online form to include the third-party contact details: “The data underlying this study are publicly available from the Demographic and Health Surveys (DHS) Program. Researchers can access the Ghana 2022 DHS dataset by registering at https://dhsprogram.com and submitting a data access request. Contact: The DHS Program, ICF International, 530 Gaither Road, Suite 500, Rockville, MD 20850, USA (info@dhsprogram.com).”

Requirement 2: PLOS ONE style compliance

Please ensure the manuscript meets PLOS ONE style requirements, including title case, reference formatting, and other journal-specific guidelines.

Response: We have reviewed the manuscript against PLOS ONE formatting requirements and made the following adjustments: (a) title and short title converted to sentence case per PLOS ONE style; (b) reference list verified for completeness and formatted per Vancouver style; (c) continuous line numbers added; (d) tables embedded within manuscript body at first mention; (e) heading styles applied consistently throughout.

Requirement 3: Reference list review

Review reference list for completeness, retracted papers, and evaluate reviewer-cited references.

Response: We identified that references 6 and 17 were identical (Gebeyehu NA et al., 2022). Reference 17 has been replaced with a distinct and contextually appropriate citation: Wariri O, Edem B, Nkereuwem E, Nkereuwem OO, Umeh G, Clark E, et al. Tracking coverage, dropout and multidimensional equity gaps in immunisation systems in West Africa, 2000–2017. BMJ Glob Health. 2019;4(5):e001713. All in-text citations have been updated accordingly. No retracted papers were identified in the reference list.

ADDITIONAL CORRECTIONS

In addition to addressing the above comments, we made the following corrections identified during the revision process:

1. Numerical updates: All estimates throughout the manuscript (Abstract, Results, Discussion, Conclusions) have been updated to reflect the corrected analysis. Key changes include: full immunization 63.9% → 69.5%; partial immunization 29.3%/31.2% → 25.7% (resolving a prior abstract–table inconsistency); MR1→MR2 dropout 37.5% → 28.9% (age-corrected); concentration index 0.061 → 0.045; regional gap 32.0pp → 35.7pp; education gap 17.2pp → 18.3pp.

2. Regression coefficients updated: Adjusted odds ratios were recalculated following the revised full immunization definition: wealth rich tertile aOR 1.43 → 1.39 (95% CI: 1.04–1.87); maternal education aOR 1.35 → 1.34 (95% CI: 1.04–1.72); ANC 4+ visits aOR 1.64 → 1.55 (95% CI: 1.19–2.01); facility delivery aOR 1.81 → 1.43 (95% CI: 1.08–1.90). Media exposure and distance to facility were no longer statistically significant in the fully adjusted model.

3. R version updated: The Methods section now correctly states R version 4.5.1 (previously listed as 4.3.1).

We believe these revisions address all concerns raised by the Editor and Reviewer. The manuscript has been substantially strengthened by the correction of the MR2 age-eligibility issue, the addition of clearer methodological descriptions, and the expanded discussion of regional determinants and future research priorities. We are grateful for the opportunity to revise this manuscript and hope it is now suitable for publication in PLOS ONE.

Sincerely,

Ahlam Tunteeya Saani, BDS, MPH

Goldfield Edem Azumah, MPH

Corresponding author: Ahlam Tunteeya Saani, University of Ghana Dental School, College of Health Sciences, University of Ghana. Email: ahlamsaani@icloud.com

---

## [Decision Letter · Decision Letter 1]

25 Mar 2026

PONE-D-25-59990R1Sociodemographic and geographic determinants of childhood immunization coverage and equity in Ghana: analysis of the 2022 demographic and health surveyPLOS One

Dear Dr. Saani,

Thank you for submitting your manuscript to PLOS ONE. After careful consideration, we feel that it has merit but does not fully meet PLOS ONE’s publication criteria as it currently stands. Therefore, we invite you to submit a revised version of the manuscript that addresses the points raised during the review process.

Thank you for your careful attention to the Reviewer’s comments, and for the improved version of this paper.

I have one suggestion.  Since the reader did not see your original erroneous calculations, it is not necessary for you to ‘confess’ or apologize.  (lines 211-212 ‘which erroneously included  … ’ can simply be changed to give the correct value and to state that it is adjusted for the age-eligible population.  Similarly in the abstract and in Lines 307 and following, just explain clearly what you did without discussing your previous error.  To make this entirely clear, you might footnote the value in the relevant Table(s) and state it is adjusted for age.  Make sure that the number of participants between 12 and 18 months is clear in the manuscript.

Was there enough age-related loss of data to affect the calculation relative to later doses of Pentavalent?

A letter that responds to each point raised by the academic editor and reviewer(s). You should upload this letter as a separate file labeled ‘Response to Reviewers’.A marked-up copy of your manuscript that highlights changes made to the original version. You should upload this as a separate file labeled ‘Revised Manuscript with Track Changes’.An unmarked version of your revised paper without tracked changes. You should upload this as a separate file labeled ‘Manuscript’.

As the corresponding author, your ORCID iD is verified in the submission system and will appear in the published article. PLOS supports the use of ORCID, and we encourage all coauthors to register for an ORCID iD and use it as well. Please encourage your coauthors to verify their ORCID iD within the submission system before final acceptance, as unverified ORCID iDs will not appear in the published article. *Only* the individual author can complete the verification step; PLOS staff the individual author can complete the verification step; PLOS staff the individual author can complete the verification step; PLOS staff the individual author can complete the verification step; PLOS staff *cannot* verify ORCID iDs on behalf of authors.verify ORCID iDs on behalf of authors.verify ORCID iDs on behalf of authors.verify ORCID iDs on behalf of authors.

We look forward to receiving your revised manuscript.

Kind regards,

Patricia Evelyn Fast, MD, Ph.D.

Academic Editor

PLOS One

Journal Requirements:

Additional Editor Comments:

Thank you for your careful attention to the Reviewer’s comments, and for the improved version of this paper.

I have one suggestion. Since the reader did not see your original erroneous calculations, it is not necessary for you to ‘confess’ or apologize. (lines 211-212 which erroneously included ... can simply be changed to give the correct value and to state that it is adjusted for the age-eligible population. Similarly in the abstract and in Lines 307 and following, just explain clearly what you did without discussing your previous error. To make this entirely clear, you might footnote the value in the relevant Table(s).

Was there enough age-related dropout to affect the later doses of Pentavalent?

Reviewers’ comments:

Reviewer’s Responses to Questions

**Comments to the Author**

1. If the authors have adequately addressed your comments raised in a previous round of review and you feel that this manuscript is now acceptable for publication, you may indicate that here to bypass the “Comments to the Author” section, enter your conflict of interest statement in the “Confidential to Editor” section, and submit your “Accept” recommendation.

Reviewer #1: All comments have been addressed

2. Is the manuscript technically sound, and do the data support the conclusions?

Reviewer #1: Yes

3. Has the statistical analysis been performed appropriately and rigorously? 

Reviewer #1: Yes

4. Have the authors made all data underlying the findings in their manuscript fully available?

The PLOS Data policy requires authors to make all data underlying the findings described in their manuscript fully available without restriction, with rare exception (please refer to the Data Availability Statement in the manuscript PDF file). The data should be provided as part of the manuscript or its supporting information, or deposited to a public repository. For example, in addition to summary statistics, the data points behind means, medians and variance measures should be available. If there are restrictions on publicly sharing data—e.g. participant privacy or use of data from a third party—those must be specified.requires authors to make all data underlying the findings described in their manuscript fully available without restriction, with rare exception (please refer to the Data Availability Statement in the manuscript PDF file). The data should be provided as part of the manuscript or its supporting information, or deposited to a public repository. For example, in addition to summary statistics, the data points behind means, medians and variance measures should be available. If there are restrictions on publicly sharing data—e.g. participant privacy or use of data from a third party—those must be specified.requires authors to make all data underlying the findings described in their manuscript fully available without restriction, with rare exception (please refer to the Data Availability Statement in the manuscript PDF file). The data should be provided as part of the manuscript or its supporting information, or deposited to a public repository. For example, in addition to summary statistics, the data points behind means, medians and variance measures should be available. If there are restrictions on publicly sharing data—e.g. participant privacy or use of data from a third party—those must be specified.requires authors to make all data underlying the findings described in their manuscript fully available without restriction, with rare exception (please refer to the Data Availability Statement in the manuscript PDF file). The data should be provided as part of the manuscript or its supporting information, or deposited to a public repository. For example, in addition to summary statistics, the data points behind means, medians and variance measures should be available. If there are restrictions on publicly sharing data—e.g. participant privacy or use of data from a third party—those must be specified.

Reviewer #1: Yes

5. Is the manuscript presented in an intelligible fashion and written in standard English?

Reviewer #1: Yes

6. Review Comments to the Author

Reviewer #1: The author’s did an excellent job in the revision of this manuscript. The data presented are particularly pertinent in the current “anti-vax” climate and should provide a useful roadmap for Ghanaian Health Authorities to proceed with addressing the gaps in vaccine coverage across the country.

7. PLOS authors have the option to publish the peer review history of their article (what does this mean?). If published, this will include your full peer review and any attached files.). If published, this will include your full peer review and any attached files.). If published, this will include your full peer review and any attached files.). If published, this will include your full peer review and any attached files.

...

Reviewer #1: **Yes:** Kathleen A. Walker, PhDKathleen A. Walker, PhDKathleen A. Walker, PhDKathleen A. Walker, PhD

---

## [Author Response · Author response to Decision Letter 2]

27 Mar 2026

Dear Dr. Fast,

Thank you for your continued engagement with our manuscript (PONE-D-25-59990R1), “Sociodemographic and geographic determinants of childhood immunization coverage and equity in Ghana: analysis of the 2022 demographic and health survey.” We are grateful for the editor’s constructive feedback and are pleased to submit our second revised manuscript for your consideration.

We are delighted that Reviewer 1 has accepted all revisions from the previous round. In response to the editor’s remaining comments, we have made the following targeted changes:

1. Revised language throughout the manuscript. We have removed all apologetic or confessional phrasing referring to earlier calculations. The corrected dropout rate is now presented straightforwardly as an age-adjusted estimate. Specifically, we have revised the relevant passages in the Results (Lines 177–179), and Discussion (Lines 247–248 and 290–292) accordingly.

2. Added footnotes to the relevant tables. We have added footnotes to Table 2 (Individual Vaccine Coverage) and Table 4 (Dropout Rates) to clearly distinguish the age-adjusted estimates from the unadjusted figures and to make the basis of each calculation transparent to readers.

3. Clarified the 12–17 month sample size. We have added a sentence to the Study Population section of the Methods explicitly stating that 1,144 children (30.2%) were aged 12–17 months and were therefore not yet age-eligible for MR2, and that MR2-specific analyses were restricted to the 2,644 children aged 18–35 months.

4. Addressed the Pentavalent question. We have confirmed in both the Response letter and the Methods section that the 12–17 month age group did not introduce any age-related data loss affecting the Pentavalent analysis. As the Pentavalent series is scheduled at 6, 10, and 14 weeks of age, all 3,788 children in the analytic sample had well exceeded eligibility age for Penta3 by the time of survey. The reported 10.4% Penta1-to-Penta3 dropout therefore reflects genuine non-completion of the series.

We believe the manuscript is now ready for publication and hope the revisions meet with your approval. Please do not hesitate to contact us if any further clarification is needed.

Yours sincerely,

Goldfield Edem Azumah

---

## [Editor Report · Decision Letter 2]

1 Apr 2026

Sociodemographic and geographic determinants of childhood immunization coverage and equity in Ghana: analysis of the 2022 demographic and health survey

PONE-D-25-59990R2

Dear Dr. Saani,

We’re pleased to inform you that your manuscript has been judged scientifically suitable for publication and will be formally accepted for publication once it meets all outstanding technical requirements.

An invoice will be generated when your article is formally accepted. Please note, if your institution has a publishing partnership with PLOS and your article meets the relevant criteria, all or part of your publication costs will be covered. Please make sure your user information is up-to-date by logging into Editorial Manager at Editorial Manager® and clicking the ‘Update My Information’ link at the top of the page. For questions related to billing, please contact  and clicking the ‘Update My Information’ link at the top of the page. For questions related to billing, please contact  and clicking the ‘Update My Information’ link at the top of the page. For questions related to billing, please contact  and clicking the ‘Update My Information’ link at the top of the page. For questions related to billing, please contact billing support....

Kind regards,

Patricia Evelyn Fast, MD, Ph.D.

Academic Editor

PLOS One

Additional Editor Comments (optional):

Reviewers’ comments:

---

## [Editor Report · Acceptance letter]

PONE-D-25-59990R2

PLOS One

Dear Dr. Saani,

I’m pleased to inform you that your manuscript has been deemed suitable for publication in PLOS One. Congratulations! Your manuscript is now being handed over to our production team.

Lastly, if your institution or institutions have a press office, please let them know about your upcoming paper now to help maximize its impact. If they’ll be preparing press materials, please inform our press team within the next 48 hours. Your manuscript will remain under strict press embargo until 2 pm Eastern Time on the date of publication. For more information, please contact onepress@plos.org.

Kind regards,

on behalf of

Dr. Patricia Evelyn Fast

Academic Editor

PLOS One